# Altitude, not potential larval habitat availability, explains pronounced variation in *Plasmodium falciparum* infection prevalence in the western Kenya highlands

Colins O. Oduma[1,2], Maurice Ombok[2], Xingyuan Zhao[3], Tiffany Huwe[4], Bartholomew N. Ondigo[1,2], James W. Kazura[5], John Grieco[4], Nicole Achee[4], Fang Liu[3,4], Eric Ochomo[2], Cristian Koepfli[4] *

1 Department of Biochemistry and Molecular Biology, Egerton University, Nakuru, Kenya, 2 Kenya Medical Research Institute, Centre for Global Health Research, Kisumu, Kenya, 3 Department of Applied and Computational Mathematics and Statistics, University of Notre Dame, Notre Dame, IN, United States of America, 4 Department of Biological Sciences and Eck Institute for Global Health, University of Notre Dame, Notre Dame, IN, United States of America, 5 Case Western Reserve University, Center for Global Health and Diseases, Cleveland, OH, United States of America

* ckoepfli@ed.edu

## Abstract

Progress in malaria control has stalled over the recent years. Knowledge on main drivers of transmission explaining small-scale variation in prevalence can inform targeted control measures. We collected finger-prick blood samples from 3061 individuals irrespective of clinical symptoms in 20 clusters in Busia in western Kenya and screened for *Plasmodium falciparum* parasites using qPCR and microscopy. Clusters spanned an altitude range of 207 meters (1077–1284 m). We mapped potential mosquito larval habitats and determined their number within 250 m of a household and distances to households using ArcMap. Across all clusters, *P. falciparum* parasites were detected in 49.8% (1524/3061) of individuals by qPCR and 19.5% (596/3061) by microscopy. Across the clusters, prevalence ranged from 26% to 70% by qPCR. Three to 34 larval habitats per cluster and 0–17 habitats within a 250m radius around households were observed. Using a generalized linear mixed effect model (GLMM), a 5% decrease in the odds of getting infected per each 10m increase in altitude was observed, while the number of larval habitats and their proximity to households were not statistically significant predictors for prevalence. Kitchen located indoors, open eaves, a lower level of education of the household head, older age, and being male were significantly associated with higher prevalence. Pronounced variation in prevalence at small scales was observed and needs to be taken into account for malaria surveillance and control. Potential larval habitat frequency had no direct impact on prevalence.

## Introduction

Malaria remains a major public health concern, with sub-Saharan Africa bearing the greatest burden of the disease [1]. While small-scale heterogeneity in malaria prevalence has long been

**Data Availability Statement:** All data is provided as supplementary file.

**Funding:** This work was supported by the National Institutes of Health grant R21AI137891 to CK, and by the University of Notre Dame. The funders had no role in study design, data collection and analysis, decision to publish, or preparation of the manuscript.

**Competing interests:** The authors declare that no competing interests exist.

observed [2–8], the drivers of such heterogeneity are not well understood. Understanding where infection prevalence and transmission potential are higher is needed to develop targeted control interventions.

Asymptomatic malaria infections are prevalent across communities in all transmission settings. Transmission stemming from such infections is increasingly being recognized as a key obstacle to malaria elimination. A recent study in western Kenya estimated 95% of transmission to be stemming from asymptomatic carriers [9]. Other mosquito infectivity studies in multiple countries corroborated asymptomatic individuals to be significant drivers of transmission in the community [10, 11]. Approaches to identify and/or treat *Plasmodium* parasite infections in the community, such as focal screen-and-treat [12], mass screen-and-treat [13], seasonal chemoprevention [14], mass drug administration [15], reactive case detection [16] or a combination thereof [17] are increasingly trialed or implemented. These approaches require data on malaria prevalence and risk factors of infection to inform on their best usage to reduce transmission in the community.

In parallel to interventions aiming at reducing the asymptomatic reservoir, vector control is key to reduce malaria transmission. Data on how mosquito larval habitat occurrence impacts malaria transmission can help to design integrated vector control interventions. Mosquito larval habitats are key drivers of adult vector populations [18–20]. The dominant vectors in western Kenya are *Anopheles gambiae*, *An. arabiensis* and *An. funestus* [21]. *Anopheles gambiae*, *An. arabiensis* largely share similar requirements for the larval environment, namely habitats with shallow waters exposed to sunlight [22, 23]. *An. funestus* prefers a wide range of habitats including waterbodies that are mostly shady, permanent or semi-permanent, with floating or thick vegetation and algae [18, 24, 25].

Busia County in western Kenya experience year-round malaria transmission. The transmission peaks during and shortly after seasonal rains. Over the past 2 decades, control programs have resulted in 88% reduction in malaria prevalence at the national level [26]. However, the marked reduction in prevalence nationally is not experienced locally in western Kenya [27]. This necessitates understanding factors that continue to drive transmission in this region to help design measures to counteract the factors.

To explore fine-scale spatial variation in *Plasmodium* parasite transmission and possible explanations for differences in prevalence, we determined *P. falciparum* infections by qPCR and microscopy among over 150 individuals in each of 20 clusters in Busia and assessed individual-, household-, and cluster-level factors that may be associated with the infections. Additionally, we mapped potential larval habitats such as wetland (i.e., ditch, swamp, marsh, shallow well, sand pit or pond), forest and sugarcane plantations in the region and determined their proximity to households and their frequency within 250 m of households. The study site covered an area of approximately 236 km$^2$, and the clusters spanned an altitude gradient of 207 m.

## Materials and methods

### Study population

**Ethics statement.** This study was approved by the Kenya Medical Research Institute—Scientific and Ethics Review Unit (approval no. 3931), and the University of Notre Dame Institutional Review Board (approval no. 19-04-5321). Informed written consent was obtained from all adults prior to sample collection. Also, informed written assent was obtained from children aged 12–17 years with accompanying informed written consent from the parent or legal guardian.

**Study site.** The study was conducted in Teso South subcounty of Busia County in western Kenya between late July and August 2021, 6 weeks after the end of the long rainy season. In western Kenya, *P. falciparum* is the primary malaria parasite species [28]. The area experiences seasonal rains in March-June and September-October and an annual rainfall between 760 mm and 2000 mm. The maximum temperature ranges between 26˚C and 30˚C, and minimum temperature is between 14˚C and 22˚C. The main malaria vector population in Teso South is *An. gambiae s.s* followed by *An. arabiensis* and *An. funestus* [21].

The region has a population of approximately 168,116 persons, and a land area of approximately 236 km$^2$ and has undulating terrain intersected by wetland features including rivers, swamps, marshlands, and drainage ditches, among others. The existence of wetland features and the farmland activities promote the establishment of larval habitats. In western Kenya region, bed nets are distributed every three years since 2004 by the National Malaria Control Program [29, 30].

**Sample collection.** Clusters were defined as villages with approximately 150 households each. They were purposively selected based on their differences in altitude and designed to cover the study area evenly, mapped and separated by a minimum of 400 m (Fig 1). Convenience sampling was employed to sample households within 20 clusters. 27–59 households per cluster were sampled to reach a minimum of 150 individuals per cluster.

Household information including Global Positioning System (GPS) coordinates, altitude above sea level, level of education of household head, kitchen location, roof material, presence of screen on window, eave type and number of participants in the household were captured using a data collection tool developed using CommCare HQ [32]. Participants' clinical and socio-demographic characteristics including age, sex, period of residency in study area, previous involvement in malaria vaccine trial, history of antimalarial treatment within the past 2 weeks and 1 month, bed-net use, and presence of malaria related clinical symptoms 2 days prior to sample collection were also captured using the data collection tool. None of the participants in the study had received the RTS,S malaria vaccine that was administered to children who participated in an implementation trial in neighboring counties of western Kenya [33, 34].

300 μL of capillary blood was collected by finger-prick into EDTA microtainer tubes (Becton Dickinson, New Jersey, United States). 6 μL and 2 μL blood were used to prepare thick and thin blood smears for microscopy. Blood was stored at -20˚C until DNA extraction.

## Household data collection

**Mosquito habitat selection and sampling.** Potential mosquito breeding habitats within a 250 m radius of the households sampled were identified and their GPS captured. The chosen radius represents a potentially feasible area for targeted or reactive vector control. While we are not aware of trials of focal larval source management, other reactive vector control interventions applied similar geographical scales [17, 35–37]. Larval source management e.g larviciding is most effective during dry season when larval habitats are few, fixed and findable [38, 39]. Therefore, we mapped habitats during the dry season in the months of February and March 2021 to determine whether they predict ongoing transmission across small scales in the region to guide on larval source management. The habitats were classified as swamp, marshland, shallow well, drainage ditch, forest, pond or sugarcane plantation. The following definitions were used in the context of this study. "Swamp" was defined as waterlogged soil having trees and other woody plants as dominant vegetation. "Marshland" was defined as waterlogged soil having non-woody plants as dominant vegetation. "Shallow well" was defined as an enlarged surface water collection hole. "Sand pit" was defined as a water ground with

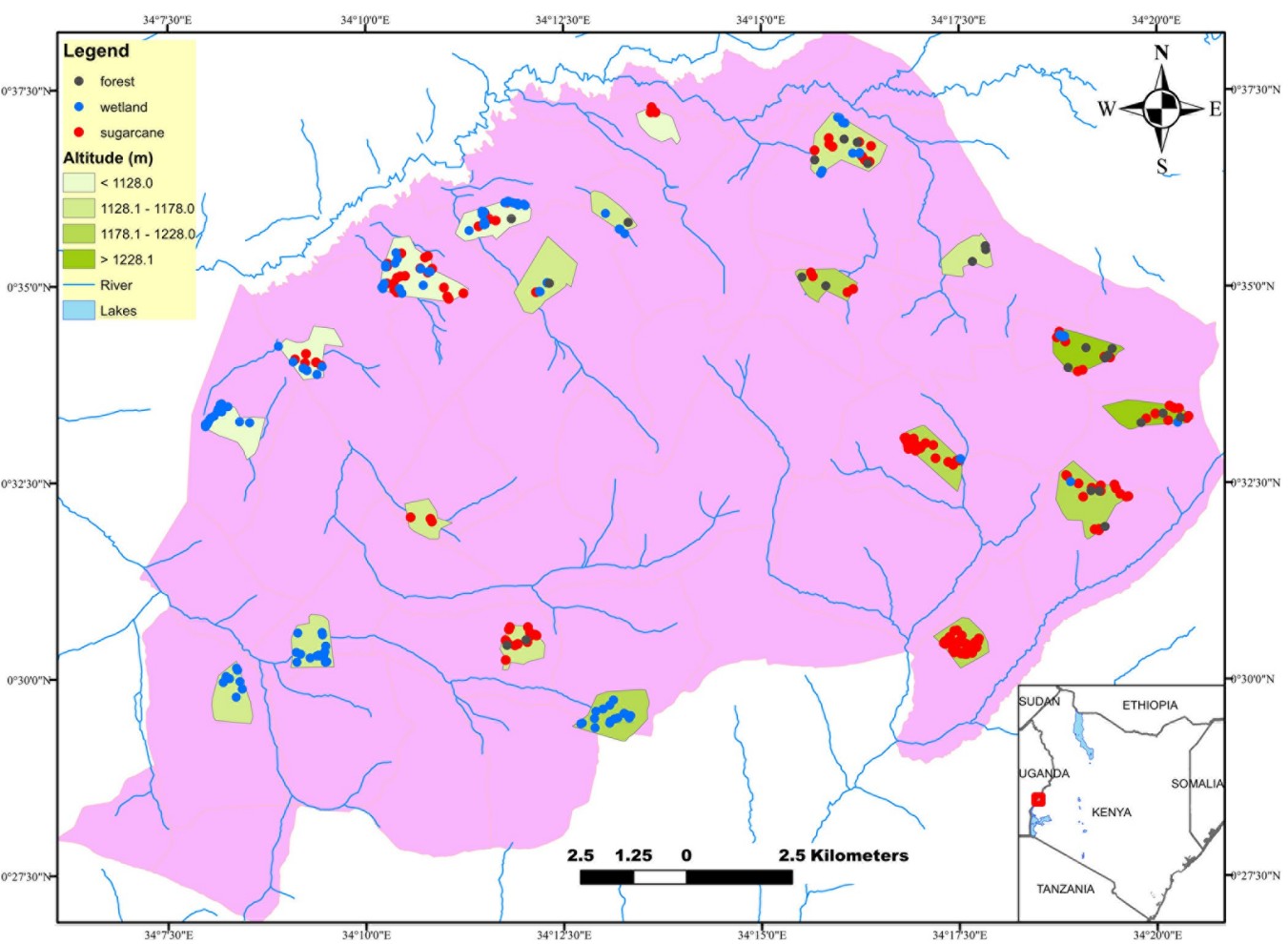

**Fig 1. Map showing study site, 20 clusters sampled, altitudinal transects and habitats distribution across the clusters.** Source of basemap shapefile [31].

excavation for sand, murram, stone or mud. "Drainage ditch" was defined as a watercourse in lower lying area off the side of a road or in farmland. "Pond" was defined as an inland body of standing water of small to medium size (approx. ≤ 50 m x 30 m). "River" was defined as natural body of water that flows downstream. "Forest" was defined using UNESCO standards as area of land > 0.5 ha with trees as dominant form of life and canopy cover comprising > 10% with a tree height of ≥5 m (Fig 2). A minimal interval of >5 m radius was considered between neighboring habitats.

**Parasite screening and quantification by microscopy.** Slides were stained according to WHO standard [47] and read by certified microscopists according to the WHO guidelines. A slide was identified as either negative or positive for *Plasmodium* species. Asexual parasite density was calculated whenever a slide was reported as positive. Each slide was read by two independent microscopists. A third independent reader was involved in case of discrepancy in identifying a positive slide, species, and whenever differences in asexual parasite density exceeded acceptable range between the readers. Parasite density was determined by averaging the densities reported by the readers. For thick film microscopy, 40 high-power fields were examined and the density of parasite determined assuming 8000 white blood cells /μL of blood. For thin film microscopy, the density was determined from parasite counts per 2000

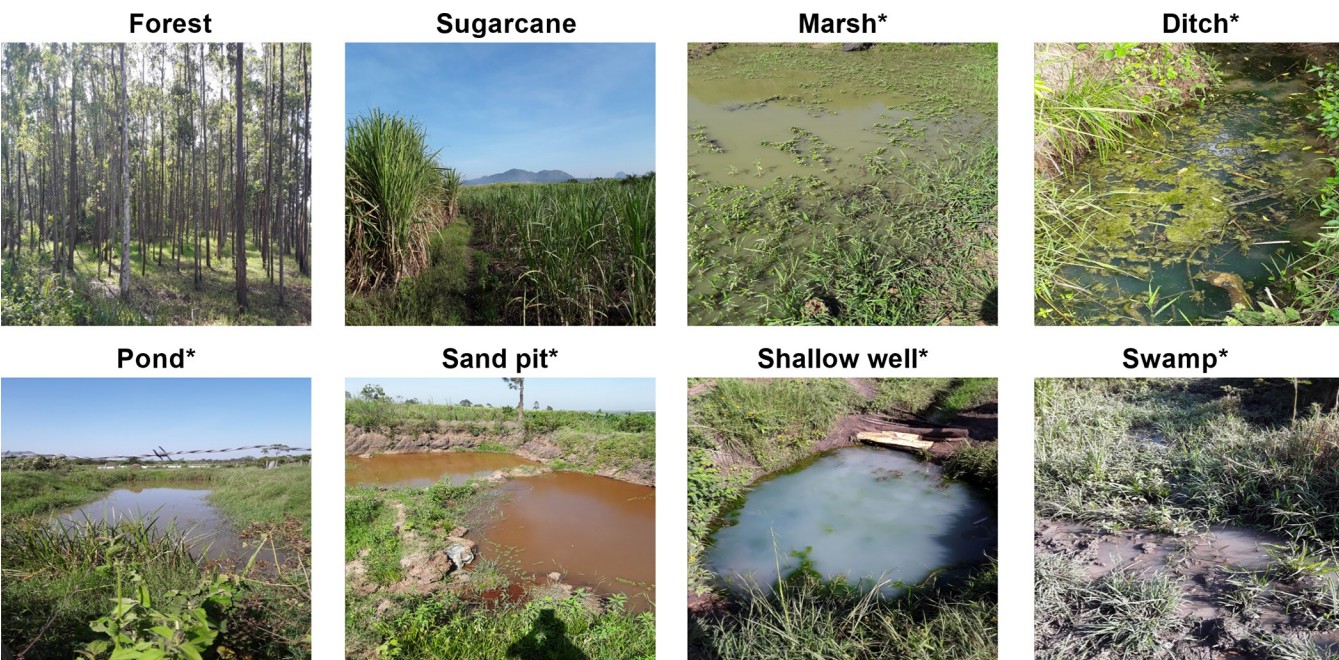

**Fig 2. Habitats identified by the present study in the dry season.** Habitats marked with asterisk (*) collectively are referred to as wetlands in this article. Elsewhere, the habitats are proxies for Anopheline larval habitats [40–46].

red blood cells assuming 5 million red blood cells/μL of blood. A slide was declared negative after examining 100 microscopic fields.

**Molecular parasite screening and quantification.** DNA was extracted from 100 μL blood using the Genomic DNA Extraction kit (Macherey-Nagel, *Düren*, Germany) and eluted in an equivalent volume of elution buffer. 4 μL of DNA, corresponding to 4 μL of blood was screened for *P. falciparum* using ultrasensitive qPCR that amplifies a conserved region of the *var* gene acidic terminal sequence (*var*ATS) according to a previously published protocol [48]. The *var*ATS gene assay amplifies ~20 copies/genome [48]. Absolute parasite densities were obtained using an external standard curve of ten-fold serial dilutions (5-steps) of 3D7 *P. falciparum* parasites quantified by droplet digital PCR (ddPCR) [49]. The ddPCR thermocycling conditions, sequences and concentration of primers and probe are given in an additional file [50].

**Statistical analysis.** Parasite densities were $\log_{10}$ transformed and geometric means per μL blood calculated whenever densities were reported. The Shapiro-Wilk test was employed to test the normality of data following log transformation. Spearman correlations at the cluster level were calculated between microscopy and qPCR prevalence, altitude and prevalence, habitat number and prevalence, and habitat number and altitude. Heterogeneity in parasite prevalence and proportion of submicroscopic infections across the clusters were determined using the $\chi^2$ test for trends in proportions.

The difference in densities between submicroscopic and microscopy positive infections among the qPCR positive data subset was estimated by a linear mixed effect model on the log-transformed parasite density data. The model contains results on microscopy status (positive vs. negative) as a fixed effect and two random effects of cluster and household nested within cluster. The geometric means (95% confidence intervals (CIs)) of parasite density were obtained for both groups, as well as their difference, along with a Wald-type test. Differences in densities across age-groups were estimated by a linear mixed effect model on the log-

transformed parasite density data. The model contains age as a fixed effect and two random effects of cluster and household nested within cluster. The geometric means (95% CIs) of parasite density were obtained for each age group. Study map graphics and determination of distance from the households to the nearest river or other habitats were done in ArcMap version 10.3. Statistical analysis was performed in R version 4.0.2 and GraphPad Prism version 9.0.

A generalized linear mixed effect model (GLMM) with the logit link was used to analyze the dichotomous data on malaria infection detection by PCR and microscopy, respectively. The model is formulated as logit $(\mu) = \log (\mu/(1-\mu)) = X\beta + Zb$, where $\beta$ contains the unknown fixed effects with corresponding design matrix $X$, $b$ contains the unknown random effects with corresponding design matrix $Z$, and $\mu$ is the probability of malaria infection given $X$ and $Z$. $X$ includes age category, sex, altitude, education level of household head, kitchen location, roof material, screens window, eave type, household population, bed net coverage percentage, distance to the nearest river, distance to the nearest habitat within cluster, habitat number per cluster, and habitat number within 250 m of household as fixed effects, and $Z$ includes two random effects of cluster and household nested within cluster. The following numerical attributes were standardized before model fitting: altitude, household population, distance to the nearest river, distance to the nearest habitat within cluster, habitat number per cluster, and habitat number within 250 m of household. The results were transformed back to the original scale for reporting.

The maximum likelihood estimates exp $(\hat{\beta})$ for the odds ratio (OR) of malaria infection associated with each covariate was obtained, along with the 95% CI, via the Laplace approximation of the marginal likelihood function. The raw *p*-value on the effect of each covariate was calculated, as well as the FDR (False Discovery Rate)-adjusted *p*-value (or *q*-value). The FDR-adjusted *p*-value is calculated by $q_{(i)} = \min_{k=i,\ldots m}\{\min(p_{(i)}m/k, 1)\}$ for $i = 1,\ldots,m$, where $p_{(i)}$ is the *i*-th smallest *p*-value among the *m* *p*-values associated with *m* multiple tests.

From the GLMM, we can estimate the clustering effects at the cluster and household within cluster levels, respectively, after adjusting for the model covariates *X*. We used the median odds ratio (OR) method [51] to estimate the clustering effects, and median OR $\geq 1$. When there is no clustering effect, median OR is 1; the further away median OR is from 1, the stronger the clustering effect. Specifically, the median OR for cluster in this study is $\exp(\sqrt{2}\sigma_1\Phi^{-1}(0.75))$ and the median OR for household nested within cluster is $\exp(\sqrt{2}\sigma_2\Phi^{-1}(0.75))$, where $\sigma_1^2$ is the variance of the random effect of cluster and $\sigma_2^2$ is the variance of the random effect of household, respectively.

## Results

### Prevalence and density of *P. falciparum* infections

3061 participants across 20 clusters aged between 1 year to 99 years, representing the age distribution of the general population, were sampled. Baseline covariates are shown in Table 1. The majority of the study population, (94.9%, 2906/3061) did not report clinical symptoms of malaria within two days prior to blood draw. Among the 155 participants harboring malaria related clinical symptoms (i.e., symptomatic individuals), 22 (14.2%) were positive by microscopy and 64 (41.3%) by qPCR, while 69 (44.5%) did not carry any detectable parasites. The proportion of individuals reporting symptoms did not differ among age groups (*P* = 0.907). Most of the participants (92.9%, 2845/3061) reported using a bed net.

Across all clusters, *P. falciparum* parasites were detected in 19.5% (596/3061) of individuals by microscopy and in 49.8% (1524/3061) of individuals by qPCR. Submicroscopic infections (i.e., infections detected by qPCR but not by microscopy) accounted for 60.9% (928/1524) of

**Table 1. Summary statistics of baseline covariates.**

| | Numerical variable | | | | | |
|---|---|---|---|---|---|---|
| | variable | min | max | median | mean | std dev |
| Household level (n = 746) | Altitude | 1077 | 1284 | 1149 | 1162 | 48.5 |
| | Distance to the nearest river | 0.4 | 1933.6 | 510.5 | 581.8 | 426.9 |
| | Household population | 1 | 16 | 5 | 5.5 | 2.8 |
| | Habitat number within 250m of household | 0 | 17 | 2 | 2.9 | 2.9 |
| | Distance to the nearest habitat within cluster | 0.5 | 881.3 | 127.1 | 160.9 | 128.3 |
| Cluster level (n = 20) | Bed net coverage % | 73.9% | 100% | 93.5% | 92.9% | 6.2% |
| | Habitat number per cluster | 3 | 34 | 15 | 15.5 | 9.2 |
| | Categorical variable | | | | | |
| | variable | category | | | percentage | frequency |
| Household level (n = 746) | Education of household head | Below secondary | | | 76.5% | 2343 |
| | | Completed secondary | | | 23.5% | 718 |
| | Kitchen location | Indoor | | | 19.6% | 599 |
| | | Outdoor | | | 80.4% | 2462 |
| | Roof material | Corrugated iron | | | 84.4% | 2583 |
| | | Non corrugated iron | | | 15.6% | 478 |
| | Screens window | No | | | 97.2% | 2976 |
| | | Yes | | | 2.8% | 85 |
| | Eave type | Closed | | | 32.7% | 1002 |
| | | Open | | | 67.3% | 2059 |
| Individual level (n = 3061) | Age category | <5 years | | | 16.8% | 514 |
| | | 5–15 years | | | 25.9% | 794 |
| | | >15 years | | | 57.3% | 1753 |
| | Sex | Female | | | 59.5% | 1822 |
| | | Male | | | 40.5% | 1239 |

the infections. The geometric mean parasite density by qPCR was 34.1 parasites/µL, with a 95% confidence interval (CI95) of 24.8 to 46.8 parasites/µL. Densities in submicroscopic infections (4.6 parasites/µL (CI95: 3.7, 5.7)) were statistically significantly lower ($P<0.001$) than those in microscopy-positive and PCR-positive infections (1018.2 parasites/µL (CI95: 794.5, 1304.9)).

Across all clusters, 746 households were sampled, with 1 to 16 individuals per household (median = 5). The majority of households (98.3%, 733/746) had at least one bed net. 77.2% (576/746) of the households had at least one *P. falciparum* infected individual by qPCR, and 45.0% (336/746) of the households had a least one infection by microscopy. Detecting an infection by microscopy in a household was not a predictor for the presence of submicroscopic infections. Among 336 households with at least one microscopy positive individual, submicroscopic infections were detected in 29.8% (514/1722) of household members. Among 410 households with no microscopy positive infections, submicroscopic infections were detected in 32.0% (429/1339) of household members.

Prevalence by qPCR peaked in children aged 5–15 years at 60.3%, and was significantly higher than those in children <5 years (39.7%) and adults >15 years (48.0%, $P<0.001$). Likewise, parasite densities by qPCR differed significantly among age groups (S1 Fig). The densities were significantly higher ($P<0.001$) in children aged 5–15 years with a geometric mean of 94.9 parasites/µL (CI95:63.5, 141.9), which was 4.5 folds of that in adults >15 years (21.2 parasites/µL (CI95:15.2, 29.7)) and 3.7 folds of that in children <5 years (25.5 parasites/µL (CI95:14.8,

43.8)). As a result, the proportion of submicroscopic infections was highest in adults (70.2%) compared to children aged 5–15 years (46.8%) and <5 years (63.2%) (*P* < 0.001, S1 Fig). Among infections detected by microscopy, prevalence was highest in children aged 5–15 years (53.2%). Prevalence of infection was significantly higher in males compared to females (55.0% vs 46.2%, *P*<0.001).

## Heterogeneity in prevalence across clusters

Prevalence by microscopy and qPCR varied significantly across the clusters. By microscopy, prevalence per cluster ranged from 7.8% (12/153) to 32.7% (50/153, *P*<0.001). By qPCR, prevalence per cluster ranged from 26.1% (40/153) to 70.6% (108/153, *P*<0.001). Prevalence per cluster by microscopy and qPCR were strongly correlated (R = 0.758, *P*<0.001, S2 Fig).

The proportion of submicroscopic infections differed significantly across clusters, ranging from 47.7% to 77.1% (*P*<0.002). The proportion of submicroscopic infections and parasite prevalence by microscopy exhibited a strong and highly significant inverse correlation (R = -0.840, *P*<0.001, S2B Fig). In four clusters with highest prevalence, 47.7–53.3% of infections were submicroscopic, while in four clusters with lowest prevalence, 72.2–77.1% were submicroscopic. In contrast, parasite prevalence by qPCR and the proportion of submicroscopic infections were not correlated (*P* = 0.105, S2 Fig). The bed net usage varied across clusters (74–100%, *P*<0.001), with 16/20 clusters having over 90% of participants using bed nets.

## Association of altitude and larval habitats on infection prevalence

For a possible explanation of the pronounced differences in infection prevalence across clusters, the abundance and types of potential larval habitats were mapped. Four types of habitats were differentiated: (i) Wetland habitats i.e., water sources such as swamps, ponds, marshes, shallow wells, sand pits and ditches, (ii) rivers (the distance from each household to the closest river was calculated), (iii) forests, and (iv) sugarcane plantations (Fig 2, Table 1).

309 potential habitats were identified across the twenty clusters (Fig 1). 3 to 34 potential habitats were identified per cluster, and 13 clusters had over 10 potential habitats (Table 1). The most frequent potential habitats across all the clusters combined were sugarcane plantations (n = 145, 46.9%) and wetlands (n = 139, 45.0%). Rivers intersecting the study area were widely distributed and not clustered in specific areas (Fig 1). None to 17 habitats were identified within a 250 m radius around each house (Table 1). In univariate analysis, the number of habitats (wetland, sugarcane and forest combined) per cluster did not predict prevalence (qPCR: *P* = 0.955, microscopy: *P* = 0.951, Fig 3A and 3B). The number of habitats stratified by type per cluster also did not predict prevalence (S3 Fig).

The clusters spanned an altitude range of 207 meters, ranging from 1077 to 1284 m (Table 1). There was a moderate significant inverse correlation between prevalence per cluster by qPCR and altitude (R = -0.462, *P* = 0.040, Fig 3C), but no correlation between prevalence per cluster by microscopy and altitude (*P* = 0.444, Fig 3D).

The number of habitats (wetland, sugarcane, forest combined) per cluster did not differ with increasing altitude (*P* = 0.672, Fig 3E). Likewise, the number of each habitat type did not vary with increasing altitude (Wetland: *P* = 0.074, sugarcane: *P* = 0.373, forest: *P* = 0.093, S3 Fig).

## Identification of risk factors for infection

We used a generalized linear mixed effect model to identify risk factors associated with prevalence of infection and covariates that correlate with variation in infection prevalence across the clusters. All covariates (Table 1) were included in the model except those of extreme skewness

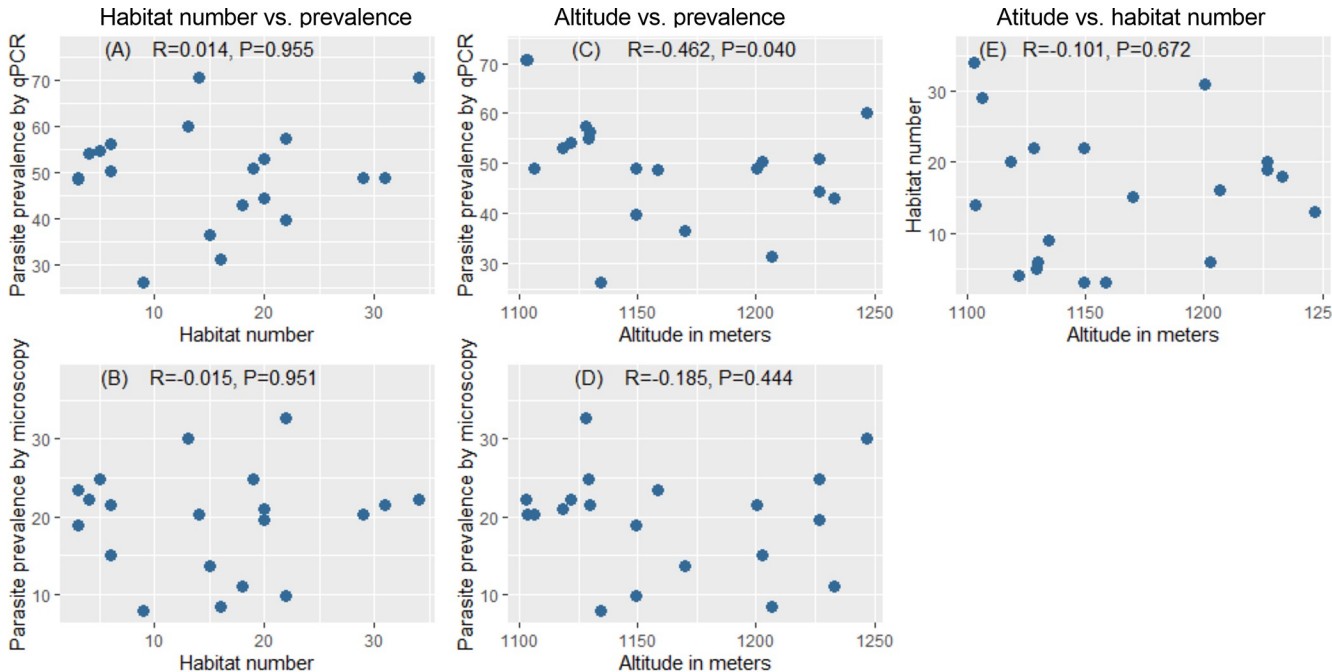

**Fig 3. Relationships among prevalence, habitat number, and altitude across clusters.** The habitats comprise forest, wetland and sugarcane combined. Each dot represents a cluster. Two clusters had identical prevalence of 70.6% by qPCR thus dots overlap. R values are calculated by spearman correlation.

that would cause difficulties in model fitting, i.e., residency, symptoms, and history of treatment.

Age, sex, eave type, education level of the head of the household, kitchen location and altitude were statistically significantly associated with the likelihood of *Plasmodium* infection by qPCR. In contrast, number of habitats per cluster, the number of habitats within a 250 m radius around the household, roof material, household population, and screens on windows had no significant impact on prevalence (Table 2). The distance to the nearest river and distance to the nearest habitat within cluster had no significant impact on prevalence (Table 2).

Every 10 m increase in altitude resulted in a 5.1% reduction in the odds of infection by qPCR. Having the kitchen outdoors resulted in a 31.0% decrease in the odds of infection by qPCR. If the head of the household completed secondary education, the odds of infection by qPCR decreased by 36.7% compared to households where the head did not attend secondary school. Having an open eave resulted in a 53.4% increase in the odds of infection by qPCR compared to houses with closed eaves (Table 2).

Similar trends in predictors of infections were observed by microscopy. School age children, males, open eaves and a low education level of head of household was associated with higher odds of infection. In contrast, altitude, the total number of larval habitats per cluster, the number of habitats within 250 m of a household, kitchen location, population size of the household and the distance to the nearest river had no significant impact on prevalence by microscopy (Table 2).

Clustering at the household and cluster level was assessed using the GLMM. The farther away a median odds ratio (OR) is from 1, the higher the odds a person would be infected with malaria if there is a positive case within the same household or the same cluster. The median OR for clustering of infections in households was 2.94 based on qPCR, and 2.44 based on microscopy. The median OR for clustering of infections in clusters was 1.60 based on qPCR,

**Table 2. Estimated effects of risk factors for *P. falciparum* infection from GLMM.**

| Variable | PCR | | | Microscopy | | |
|---|---|---|---|---|---|---|
| | OR (CI95) | *p*-value | *q-value* | OR (CI95) | *p*-value | *q-value* |
| **Individual-level factor** | | | | | | |
| Age (5–15 vs <5) | 2.533 (1.939, 3.307) | <0.001* | <0.001* | 2.712 (1.987, 3.703) | <0.001* | <0.001* |
| Age (≥15 vs <5) | 1.713 (1.355, 2.165) | <0.001* | <0.001* | 1.077 (0.800, 1.450) | 0.626 | 0.752 |
| Sex (male vs female) | 1.623 (1.366, 1.928) | <0.001* | <0.001* | 1.604 (1.310, 1.964) | <0.001* | <0.001* |
| **Household-level factor** | | | | | | |
| Eave type (open vs closed) | 1.534 (1.220,1.930) | <0.001* | <0.001* | 1.560 (1.197, 2.033) | <0.001* | 0.005* |
| Education of household head (completed secondary vs below secondary) | 0.633 (0.493, 0.813) | <0.001* | 0.001* | 0.636 (0.472, 0.857) | 0.003* | 0.011* |
| Kitchen location (Outdoor vs Indoor) | 0.690 (0.526, 0.903) | 0.007* | 0.017* | 0.720 (0.541, 0.959) | 0.025* | 0.074 |
| Altitude (every 10-unit increase) | 0.949 (0.912, 0.987) | 0.009* | 0.019* | 0.972 (0.935, 1.011) | 0.159 | 0.261 |
| Distance to the nearest river (every 100- meter increase) | 1.038 (0.997, 1.081) | 0.068 | 0.127 | 1.043 (1.002, 1.087) | 0.041* | 0.089 |
| Roof (non-corrugated iron vs corrugated iron) | 1.267 (0.941, 1.707) | 0.118 | 0.197 | 0.949 (0.685, 1.316) | 0.754 | 0.808 |
| Household population (increase by 1) | 1.031 (0.989, 1.075) | 0.147 | 0.220 | 1.048 (1.002,1.097) | 0.040* | 0.089 |
| Distance to the nearest habitat within cluster (every 100-metre increase) | 0.929 (0.829, 1.042) | 0.210 | 0.263 | 1.002 (0.885, 1.135) | 0.975 | 0.974 |
| Habitat number within 250m of household (increase by 1) | 1.028 (0.975, 1.085) | 0.307 | 0.355 | 1.041 (0.982,1.103) | 0.174 | 0.261 |
| Screens window (Yes vs No) | 0.964 (0.515,1.803) | 0.907 | 0.972 | 0.537 (0.226, 1.273) | 0.158 | 0.261 |
| **Cluster-level factor** | | | | | | |
| Bednet (coverage % per cluster) | 0.144 (0.007, 2.898) | 0.206 | 0.263 | 0.258 (0.015, 4.403) | 0.350 | 0.477 |
| Habitat number per cluster (increase by 1) | 1.000 (0.978, 1.022) | 0.984 | 0.984 | 0.995 (0.975, 1.016) | 0.651 | 0.752 |

OR: odds ratio. CI95: 95% confidence interval. Asterisk (*) indicates significant at 5% level. FDR: False Discovery Rate. *q*-value is the FDR adjusted *p*-value.

and 1.51 based on microscopy. The median OR results suggest that there is clustering at both household and cluster levels, and that clustering across households is stronger than across clusters.

## Discussion

In Busia County in western Kenya, the prevalence of *P. falciparum* infection by qPCR differed nearly 3-fold across 20 clusters. Overall, asymptomatic prevalence was high at 50%. Altitude was a key predictor for prevalence of infection, with a 5% decrease in the odds of infection per every 10 meters increase in altitude. Three to 34 potential larval habitats were found per cluster, and none to 17 habitats within a 250 m radius around households. Yet, habitat number had no impact on prevalence.

While habitat density did not predict prevalence, a strong correlation of prevalence with altitude was observed within a narrow altitude range of 207 meters. This finding corroborates previous studies across much larger altitude ranges, where significantly fewer clinical cases, lower prevalence, and reductions in the proportion of *P. falciparum* infected mosquitoes and *Anopheles* vector abundance were observed with increasing altitude [3, 4, 52–60]. Increasing altitude might result in changes in land use and fewer larval habitats. Yet, in the current study, this was not observed. Alternatively, lower temperatures at higher altitudes, which directly impact development and survival of mosquitos and *Plasmodium* parasites, might cause the observed pattern [61].

Additional individual and household factors were determinants of heterogeneity in parasite transmission across the clusters. Infection prevalence was higher among individuals living in households with open eaves and with the kitchen located indoors. The impact of kitchen location might be explained by local vector population behavior. The main mosquito vectors in

this region are *Anopheles gambiae s.s.* and *An. funestus* [21] that exhibit endophilic and endophagic behaviors [62, 63]. Spending more time indoors thus increases the risk of infection. The endophagic behavior of the mosquitos can also explain the impact of open eaves, which was associated with increased risk of infection. Prevalence was also higher among males, in school age children, and in households where the head had not completed secondary school, corroborating previous findings [64–69].

Prevalence by microscopy was a good predictor for submicroscopic prevalence at the cluster level. Thus, screening by microscopy might inform interventions to reduce the asymptomatic prevalence, e.g. through targeted mass drug administration. Clustering of infections was also observed at the household level. Yet, the presence of microscopy-positive infections was not a predictor for submicroscopic infections households, and thus cannot guide control strategies towards further targeted treatment. This contrasts with observations in other sites [70] and with numerous studies where prevalence of asymptomatic and/or submicroscopic infections was higher in household members of clinical cases, a pattern exploited by reactive case detection programs [16, 36, 71–73]. The contrasting findings might be explained by differences in transmission intensity. Reactive case detection is often used in low transmission or elimination areas where clustering of infections becomes more pronounced [74, 75], while the current study was carried out in a moderate-to-high transmission area.

The relationship between proximity to potential larval habitats and risk of infection is complex. Some studies have reported an increased prevalence or incidence among residents in close proximity to larval habitats [4, 71, 76–79], while others have not [80, 81]. Our habitat mapping had important limitations. Not all potential habitats are equally suited for larval development. We did not assess whether potential habitats carried larvae. Multiple environmental factors including temperature, light, salinity, vegetation, hydrology and geomorphology determine the establishment of larval habitats [82]. Yet the types of habitats identified in the present study were found elsewhere to contain mosquito larvae, e.g. drainage ditches, forests, and swamps [44, 46, 80, 83, 84]. Further, we did not map very small and unstable potential habitats, such as small buckets or hoof prints that could have an impact on transmission [24, 85]. Analysis of the impact of proximity to rivers was hampered by low variation in this parameter, as most of the clusters were evenly transected by rivers. Because of their low numbers, different types of water sources (ditch, swamp, marsh, shallow well, sand pit, and pond) were grouped together when analyzing their association with prevalence.

As a further limitation, habitat density and prevalence were determined at only one time point each. It is possible that habitat density predicts transmission intensity at the beginning of the rainy season, but that this effect wanes until the end of the main transmission season, when samples for the current study were collected. Of note, a previous study in western Kenya found moderate differences in prevalence between the dry and the wet season (17.5% vs. 13.5% by qPCR), i.e. less pronounced variation than among clusters in the present study [50]. Further research is needed to determine whether small-scale spatial variation in prevalence is maintained throughout seasons, and whether habitat density and prevalence correlate during some times of the year only.

Overall, half of the population sampled carried parasites. This reservoir presents a major roadblock for improved malaria control and elimination. In addition to improved vector control, programs to actively clear these infections from the population might be needed. Nearly two thirds of infections were submicroscopic, and thus might not be detected by mass screen-and-treat campaigns. A novel generation of highly sensitive rapid diagnostic tests detect ultra-low density infections [86, 87]. Whether this increased sensitivity is sufficient to reduce transmission if used in mass test and treatment or reactive case detection campaigns remains to be shown.

In conclusion, pronounced variation in asymptomatic prevalence at small scales needs to be considered for malaria surveillance and the evaluation of control efforts in western Kenya. Given the absence of a direct impact of potential larval habitat density on prevalence at small scales, larval source management will need to target all potential habitats across a wider geographical area.

## Supporting information

**S1 Fig.** Age trends in *P. falciparum* density (A) and prevalence (B) by qPCR, and proportion of submicroscopic infections. Error bars in panel A shows standard errors of the geometric mean. In panel B, the proportion of submicroscopic infections is shown in addition to prevalence.
(TIF)

**S2 Fig. Relationships between prevalence by microscopy and qPCR, and proportion of submicroscopic infection across clusters.** Each dot represents a cluster. R values are calculated by spearman correlation.
(TIF)

**S3 Fig. Relationship among habitat type number, prevalence, and altitude across clusters.**
(TIF)

**S1 Data. Database used for analysis.**
(XLSX)

## Acknowledgments

We are grateful to all study participants. Further, we acknowledge the technical support from Brian Polo in building the study CommCare data collection tool. This work is published with the permission of the KEMRI Director General.

## Author Contributions

**Conceptualization:** Colins O. Oduma, John Grieco, Nicole Achee, Eric Ochomo, Cristian Koepfli.

**Formal analysis:** Xingyuan Zhao, Fang Liu.

**Funding acquisition:** Cristian Koepfli.

**Investigation:** Colins O. Oduma, Eric Ochomo, Cristian Koepfli.

**Methodology:** Colins O. Oduma.

**Project administration:** Tiffany Huwe, Eric Ochomo, Cristian Koepfli.

**Resources:** Eric Ochomo.

**Supervision:** Bartholomew N. Ondigo, James W. Kazura, Fang Liu, Cristian Koepfli.

**Validation:** Fang Liu.

**Visualization:** Colins O. Oduma, Maurice Ombok.

**Writing – original draft:** Colins O. Oduma, Cristian Koepfli.

**Writing – review & editing:** Colins O. Oduma, James W. Kazura, Fang Liu, Eric Ochomo, Cristian Koepfli.

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
