## [Decision Letter · Decision Letter 0]

11 Jan 2023

PGPH-D-22-02053

Altitude, not potential larval habitat availability, explains pronounced variation in Plasmodium falciparum infection prevalence in the western Kenya highlands

Dear Dr. Koepfli,

Thank you for submitting your manuscript to PLOS Global Public Health. After careful consideration, we feel that it has merit but does not fully meet PLOS Global Public Health’s publication criteria as it currently stands. Therefore, we invite you to submit a revised version of the manuscript that addresses the points raised during the review process.

We look forward to receiving your revised manuscript.

Kind regards,

Abhinav Sinha, M.D.

Academic Editor

Journal Requirements:

a. State what role the funders took in the study. If the funders had no role in your study, please state: “The funders had no role in study design, data collection and analysis, decision to publish, or preparation of the manuscript.”

b. If any authors received a salary from any of your funders, please state which authors and which funders.

3. Figure 2: Please confirm (a) that you are the photographer; or (b) provide written permission from the photographer to publish the photo(s) under our CC-BY 4.0 license.

4. Fig 1: please (a) provide a direct link to the base layer of the map (i.e., the country or region border shape) and ensure this is also included in the figure legend; and (b) provide a link to the terms of use / license information for the base layer image or shapefile. We cannot publish proprietary or copyrighted maps (e.g. Google Maps, Mapquest) and the terms of use for your map base layer must be compatible with our CC-BY 4.0 license. 

Additional Editor Comments (if provided):

Reviewers' comments:

Reviewer's Responses to Questions

**Comments to the Author**

1. Does this manuscript meet PLOS Global Public Health’s publication criteria? Is the manuscript technically sound, and do the data support the conclusions? The manuscript must describe methodologically and ethically rigorous research with conclusions that are appropriately drawn based on the data presented.

Reviewer #1: Yes

Reviewer #2: Yes

2. Has the statistical analysis been performed appropriately and rigorously?

Reviewer #1: Yes

Reviewer #2: Yes

3. Have the authors made all data underlying the findings in their manuscript fully available (please refer to the Data Availability Statement at the start of the manuscript PDF file)?

Reviewer #1: Yes

Reviewer #2: Yes

4. Is the manuscript presented in an intelligible fashion and written in standard English?

Reviewer #1: Yes

Reviewer #2: Yes

5. Review Comments to the Author

Reviewer #1: In this fine manuscript, Oduma and colleagues show that the number of potential breeding sites per village and next (<250 m) to households does not predict malaria prevalence measured during the dry season in the Western Kenya highlands. They show, however, that village altitute is a significant predictor of malaria prevalence in these same populations.

As the authors acknowledge, the main limitation of this study is that "potential" larval habitats have noot been screened for the actual presence of anopheline larvae. In other works, the negative results may be associated, at least in part, with the unsurprising difficulty in accurately identifying larval habitats in the study site.

Another limitation arises from the cross-sectional nature of the study. Both exposure (e.g., proximity to larval habitats) and outcomes (presence of malaria parasites detected by microscopy or PCR) were measured at a single time point. The reader may wonder whether larval habitat availability during the dry season might predict, for example, malaria transmission in the beginning of the next rainy season. The reader may also wonder whether the proportion of submicroascopic and asymptomatic infections found during the dry season would be similar during the rainy season. Some further discussion of these limitatons might be welcome.

The authors have collected information regarding the presence of malaria related clinical symptoms 2 days prior to sample

collection (lines 146-147), but I was not able to find any description of the proportion of infections that were symptomatic and how this changes with age and parasite density. I understand that the symptoms covariate was not included in adjusted models that aimed to identify correlates of infection, but at least a brief description of these findings might be useful.

There are a few additional minor points:

1. The authors say that "qPCR results were converted to varATS copies/μL using an external standard curve of ten-fold serial dilutions (5-steps) of 3D7 P. falciparum parasites quantified by droplet digital PCR (ddPCR)" (lines 199-201), but actually they analyze numbers of parasites (not of varATS copies) per microliter (e.g., lines 270-274). Did they simply assumed that each parasite genome has 20 varATS copies?

2. Why have the authors used the threshold of 250 m to define the presence of larval habitats next to households?

3. The following sentence is unclear: "The median odds OR for households was 2.94 based on the qPCR-detected infections

and 2.44 based on the microscopy-detected infections" (lines 380-381). Mosr likely, the authors would like to say: "The median odds ratio (OR) for households...". However, only in the next sentence the authors explain what they mean by "OR for households"" "The farther away a median OR is from 1, the higher the odds a person would be infected with malaria if there is a positive case within the same household or the same cluster." As this is a quite interesting result, a clearer description would be quite useful.

4. "S1 text" legend is: "S1 text. Age trends in P. falciparum density and prevalence by qPCR, and proportion of submicroscopic infections. Submicroscopic infections were determined as infections detected by qPCR but not by microscopy.". However, this is a supplementary figure, not a text. The legend does not describe what error bars show in panel A (standard deviation? standard errors of the geometric mean?). Panel B is unclear to me. Blue bar segments show prevalences (%) by age, but I do not understand what the black bar segments show. I believe that the authors would like to show the proportion of infections (within the blue bars) that are submicroscopic; if this is correct, they should provide a stacked chart where the total bar height indicates malaria prevalence in each age group (between 40 and 60%) and the highlighted bar segment (e.g., black) shows the prevalence of subpatent infections in each age group.

Reviewer #2: The MS is good as it has good information on the malaria cases that are unnoticed and sub clinical and submicroscopic this might give an idea of case burden they may modify the title as the main finding seems to be different than the altitudes

In Methods :

The study is only six month duration that too post rainy season then how the assessment of larval habitat were correlated ?

clusters were only based on altitudes ?

As it suggests the study started during summer months it has be added in the study duration (159-162)

In Results

the positives were among mostly asymptotic?? (261-262 line)

Table 1 shows that there is very less variation in the altitude therefore there may not be good correlation between cases and altitudes as it also depends upon internal house environment authors did not analysed as they have altitude in the title of their MS. However they have correlated with the distances of breeding sites to housholds

The Figures 3 and 4 are to be explained and could be reduces

In discussion

Asymptomatic cases are very high what is strategy for treating such cases who may not visit the facility (390 line)

What were breeding sites of both species at what distance such sites are located ? (411 lines)

Local infections are possible in case indoor biting what are control efforts done at house levels by the government in such houses . and what are the process adopted by government on case report in the region any reactive surveillance occurs ? what percent is the case reporting in the region? (419-420 lines)

presence of male mosquitoes are also one of the indicator of nearby breeding habitat

As the different land forms supports different species of mosquitoes in vicinity please also provide any other species of mosquitoes encountered or found in breeding sites?(430=433 lines)

6. PLOS authors have the option to publish the peer review history of their article (what does this mean?). If published, this will include your full peer review and any attached files.

**Do you want your identity to be public for this peer review?** For information about this choice, including consent withdrawal, please see our Privacy Policy.

Reviewer #1: **Yes: **Marcelo Urbano Ferreira

Reviewer #2: **Yes: **Himmat Singh

---

## [Decision Letter · Decision Letter 1]

13 Mar 2023

Altitude, not potential larval habitat availability, explains pronounced variation in Plasmodium falciparum infection prevalence in the western Kenya highlands

PGPH-D-22-02053R1

Dear Mr. Koepfli,

We are pleased to inform you that your manuscript 'Altitude, not potential larval habitat availability, explains pronounced variation in Plasmodium falciparum infection prevalence in the western Kenya highlands' has been provisionally accepted for publication in PLOS Global Public Health.

Best regards,

Abhinav Sinha, M.D.

Academic Editor

Reviewer Comments (if any, and for reference):

Reviewer's Responses to Questions

**Comments to the Author**

1. If the authors have adequately addressed your comments raised in a previous round of review and you feel that this manuscript is now acceptable for publication, you may indicate that here to bypass the “Comments to the Author” section, enter your conflict of interest statement in the “Confidential to Editor” section, and submit your "Accept" recommendation.

Reviewer #1: All comments have been addressed

Reviewer #2: All comments have been addressed

2. Does this manuscript meet PLOS Global Public Health’s publication criteria? Is the manuscript technically sound, and do the data support the conclusions? The manuscript must describe methodologically and ethically rigorous research with conclusions that are appropriately drawn based on the data presented.

Reviewer #1: Yes

Reviewer #2: Yes

3. Has the statistical analysis been performed appropriately and rigorously?

Reviewer #1: Yes

Reviewer #2: (No Response)

4. Have the authors made all data underlying the findings in their manuscript fully available (please refer to the Data Availability Statement at the start of the manuscript PDF file)?

Reviewer #1: Yes

Reviewer #2: Yes

5. Is the manuscript presented in an intelligible fashion and written in standard English?

Reviewer #1: Yes

Reviewer #2: Yes

6. Review Comments to the Author

Reviewer #1: The authors have addressed all questions raised by the reviewer.

Reviewer #2: AUthors have answered the queries raised by reviewers however, the association with the altitude is still need better association

7. PLOS authors have the option to publish the peer review history of their article (what does this mean?). If published, this will include your full peer review and any attached files.

**Do you want your identity to be public for this peer review?** For information about this choice, including consent withdrawal, please see our Privacy Policy.

Reviewer #1: **Yes: **Marcelo Urbano Ferreira

Reviewer #2: No
